

# Ceratopogonidae (Diptera: Nematocera) of the piedmont of the Yungas forests of Tucumán: ecology and distribution

José Manuel Direni Mancini[1,2], Cecilia Adriana Veggiani-Aybar[1], Ana Denise Fuenzalida[1,3], Mercedes Sara Lizarralde de Grosso[1] and María Gabriela Quintana[1,2,3]

[1] Facultad de Ciencias Naturales e Instituto Miguel Lillo, Universidad Nacional de Tucumán, Instituto Superior de Entomología "Dr. Abraham Willink", San Miguel de Tucumán, Tucumán, Argentina
[2] Consejo Nacional de Investigaciones Científicas y Técnicas, San Miguel de Tucumán, Tucumán, Argentina
[3] Instituto Nacional de Medicina Tropical, Puerto Iguazú, Misiones, Argentina

## ABSTRACT

Within the Ceratopogonidae family, many genera transmit numerous diseases to humans and animals, while others are important pollinators of tropical crops. In the Yungas ecoregion of Argentina, previous systematic and ecological research on Ceratopogonidae focused on *Culicoides*, since they are the main transmitters of mansonelliasis in northwestern Argentina; however, few studies included the genera *Forcipomyia*, *Dasyhelea*, *Atrichopogon*, *Alluaudomyia*, *Echinohelea*, and *Bezzia*. Therefore, the objective of this study was to determine the presence and abundance of Ceratopogonidae in this region, their association with meteorological variables, and their variation in areas disturbed by human activity. Monthly collection of specimens was performed from July 2008 to July 2009 using CDC miniature light traps deployed for two consecutive days. A total of 360 specimens were collected, being the most abundant *Dasyhelea* genus (48.06%) followed by *Forcipomyia* (26.94%) and *Atrichopogon* (13.61%). Bivariate analyses showed significant differences in the abundance of the genera at different sampling sites and climatic conditions, with the summer season and El Corralito site showing the greatest abundance of specimens. Accumulated rainfall was the variable that related the most to the abundance of *Culicoides* (10.56%), while temperature was the most closely related variable to the abundance of *Forcipomyia*, *Dasyhelea*, and *Atrichopogon*.

## INTRODUCTION

The Ceratopogonidae family constitutes a much diversified and globally widespread group of Culicomorpha. At present, it is represented by 6,267 species and 111 living genera grouped in four subfamilies (Ceratopogoninae, Leptoconopinae, Forcipomiinae and Dasyheleinae) (*Borkent, 2016*).

Corresponding author
José Manuel Direni Mancini, josemdireni@gmail.com

*Austroconops* Wirth & Lee (only one australian species), *Culicoides* Latreille, *Leptoconops* Skuse and *Forcipomyia* Meigen (subgenus *Lasiohelea*) are implied in the transmission of arbovirus, parasites and protozoa that cause diseases both in humans and other animals (*Mellor, Boorman & Baylis, 2000*; *Borkent & Spinelli, 2007*; *Veggiani Aybar et al., 2010b*; *Veggiani Aybar et al., 2015*). Other genera proportion important services in ecological systems, such as *Forcipomyia* and *Dasyhelea* Kieffer, to a lesser extent *Atrichopogon* Kieffer, *Culicoides* and *Stilobezzia* Kieffer which are potential pollinators of different crops, such as cocoa (*Theobroma cacao* Linnaeus), rubber (*Hevea brasiliensis* Müller Argoviensis), and mango (*Mangifera indica* Linnaeus) in tropical regions (*Borkent & Spinelli, 2007*; *Bravo, Somarriba & Arteaga, 2011*); while some species of *Forcipomyia* and *Culicoides* are ectoparasites of insects, sucking the lymph of lepidopterans, coleopterans, odonata, phasmids, neuropterans and hemipterans (*Borkent, 2004*). In turn, *Ceratopogon* Meigen, *Bezzia* Kieffer, *Brachypogon* Kieffer, *Monohelea* Kieffer, *Serromyia* Meigen, *Stilobezzia* Kieffer, *Palpomyia* Meigen (*Bernotienė, 2006*), *Allohelea* Kieffer (*Werner & Kampen, 2010*), *Ceratoculicoides* Wirth & Ratanaworabhan (*Huerta & Borkent, 2005*), *Alluaudomyia* Kieffer and *Echinohelea* Macfie (*Borkent & Spinelli, 2007*) are predators of small flying insects of the same or smaller size. However, the relevance of this family is given by *Culicoides* genus, which is vector of the Bluetongue virus, Schmallenberg virus, Epizootic Hemorrhagic Disease virus, African Horse Sickness virus, Akabane virus and Bovine Ephemeral Fever virus, among others, affecting ovine and bovine cattle (*Mellor, Boorman & Baylis, 2000*; *Borkent, 2004*; *Carpenter et al., 2013*); and of the transmission to humans of the Oropouche virus, *Mansonella* nematode and *Leishmania* trypanosomatidae (*Mellor, Boorman & Baylis, 2000*; *Ronderos et al., 2003*; *Borkent, 2004*; *Slama et al., 2014*).

In northwestern Argentina, studies focused mainly in *Culicoides* genus due to its epidemiologic relevance as a vector of filarial *Mansonella ozzardi* (*Shelley & Coscarón, 2001*; *Veggiani Aybar et al., 2015*; *Veggiani Aybar, Dantur Juri & Zaidenberg, 2016*). These studies determined the spatio-temporal abundance of *Culicoides* spp. and the influence of meteorological variables (temperature, accumulated rainfall, humidity, etc) in their abundance, behavior and distribution, and the interactions between pathogens and vectors (*Veggiani Aybar et al., 2010b*; *Veggiani Aybar et al., 2011*; *Veggiani Aybar et al., 2012*; *Veggiani Aybar et al., 2015*); however, such aspects have not been studied in other families of Ceratopogonidae of the area. Therefore, the objective of this study was to determine the presence and abundance of the main genera of Ceratopogonidae in piedmont forests of Tucumán province, and to evaluate the effect of meteorological variables in their distribution.

## MATERIALS AND METHODS

### Characterization of the study area

The present study was performed at Juan Bautista Alberdi department (27°35′05.89″S; 65°37′11.70″W; 400 masl), Tucumán province (Fig. 1). The area belongs to the Yungas phytogeographic region, and is specifically located at the altitudinal gradient of piedmont forests.

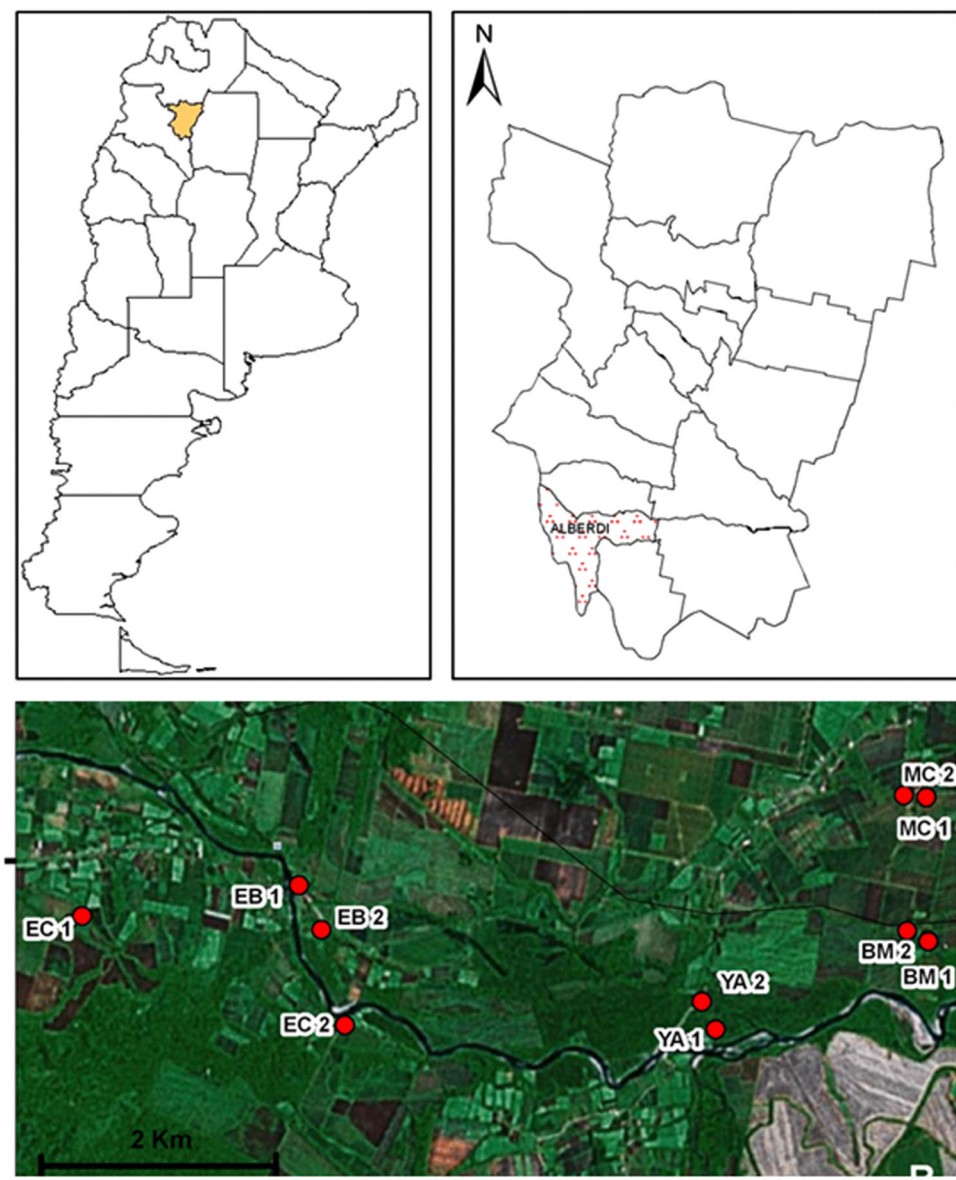

**Figure 1 Geographic distribution of sampling sites in Juan Bautista Alberdi Department, Tucumán.** EC, El Corralito; EB, El Badén; YA, Yánima; BM, Bajo Marapa; MC, Marapa Central. (from Maria G. Quintana; Date of map © 2006 Google Earth, Image © 2008 TerraMetrics).

Piedmont forests are located between 700 and 1,000 masl, and exhibit a subtropical climate, with a mean annual rainfall range of 700–1,000 mm, concentrated in summer months (November to April); a mean maximum temperature of 27.6 °C and a mean minimum temperature of 15.4 °C (*Brown & Grau, 1995*; *Malizia et al., 2012*). The native vegetation is arboreal of closed-canopy, and becomes more open near roads and at the edges of streams. Canopy vegetation is dominated by *Blepharocalyx salicifolius* (Kunth) O. Berg, *Enterolobium contortisiliquum* (Vell.) Morong, *Juglans australis* (Griseb.), and *Parapiptadenia excelsa* (Griseb.) Burkart, while undergrowth vegetation is dominated by *Piper tucumanum* C. DC., *Eugenia uniflora* L., *Urera baccifera* (L.) Gaudich.,

and *Solanum riparium* Pers. Many species of lianas of Bignoniaceae, Ulmaceae and Amarantaceae families, and vascular plants with epiphyte habits, belonging to Polipodaceae, Asplaniaceae, Piperaceae and Bromeliaceae are also frequent. In open areas, the most common arboreal species are *Tipuana tipu* (Benth.) Kuntze, *Jacaranda mimosifolia* D. Don, *Anadenanthera colubrina* var. *cebil* (Griseb.), *Tabebuia avellanedae* Lorentz ex Griseb., *Heliocarpus popayanensis* Kunth, *Fagara coco* (Gillies ex Hook. & Arn.) Engl., *Tecoma stans* (L.) Juss. ex Kunth, *Salix humboldtiana* Willd., and *Carica quercifolia* (A. St. Hil.) Hieron (*Grau, 2005*; *Brown, Malizia & Lomáscolo, 2006*).

Despite the climatic variability, rises in mean annual rainfall in the last years have been detected, as a consequence of the replacement of native vegetation and the expansion of extensive crops (sugarcane, tobacco, fruit trees, among others), which provoked important modifications in the landscape (*Brown & Malizia, 2004*).

## Collecting sites

Based on environmental and socio-demographic characteristics and operational accessibility, a total of 10 households were selected for sampling (five paired sampling sites, Fig. 1).

Peridomestic sites were georeferenced and characterized through an ad-hoc survey, using the criteria of "worst scenario," an operational definition for the site within the study area. The methodology has been employed for the study of Phlebotominae subfamily and defines sites with features such as shade presence, moist soils, organic detritus, proximity to patches of dense vegetation, density, quality and accessibility of sources of blood supply intake, no interference from external lights, and epidemiological records, among others; with higher probability of finding the specimens of interest. For spatial analyses of environmental-driven changes in the abundance of vectors, this methodology presents more biological significance than a spatial centroid (*Feliciangeli et al., 2004*; *Correa-Antonialli et al., 2007*; *Salomón, 2007*).

The following sampling sites were selected: El Corralito (EC1: 27°37′25.2″S; 65°42′59.9″W and EC2: 27°37′56.9″S; 65°41′23.9″W), El Badén (EB1: 27°37′13.9″S; 65°41′39.6″W and EB2: 27°37′27.2″S; 65°41′32.0″W), Yánima (YA1: 27°37′58.8″S; 65°39′13.7″W and YA2: 27°37′49.4″S; 65°39′19.2″W), Bajo Marapa (BM1: 27°37′30.3″S; 65°38′00.5″W and BM2: 27°37′28.4″S; 65°38′07.1″W) and Marapa Central (MC1: 27°36′46.6″S; 65°38′01.2″W and MC2: 27°36′45.1″S; 65°38′08.1″W).

## Collection and processing of specimens

Adult specimens were collected monthly from July 2008 to July 2009 with CDC-like light mini-traps (*Sudia & Chamberlain, 1962*), placed from 18:00 to 07:00 during two consecutive days. In the laboratory, specimens were separated from other insects and placed in properly labeled Eppendorf tubes containing 70% alcohol for preservation. The identification of adults was performed following the taxonomic keys of *Spinelli & Wirth (1993)* and *Spinelli et al. (2005)*.

## Data analysis

The obtained data were entered into an Excel spreadsheet and then were analyzed using InfoStat 2016e version statistical software (*Di Rienzo et al., 2016*). The abundance of each genus by season and sampling site was compared and bivariate statistical analyses (chi-squared test) were applied. In all cases, Cramer V coefficient was used to measure the association or independence among the considered variables, which takes values between 0 (weak association) to 1 (strong association). Posteriorly, multiple regression analyses (stepwise method) were performed (*Balzarini et al., 2008*). The independent variables were included in the multiple linear regression analyses but the variables that contributed the least to the explanation were eliminated one by one. All statistical tests were considered significant at $P \leq 0.05$.

Mean abundance values (dependent variable) were standardized using $\log (n + 2)$. The meteorological variables (independent variables) considered in this study were: temperature (T), accumulated rainfall (R), relative humidity (Rh), wind speed (Ws) and maximum wind speed (maxWs), which were monthly averaged. Meteorological data were obtained from the Agro-meteorology department of the Agroindustrial Experimental Station Obispo Colombres, Tucumán province.

## RESULTS

A total of 360 adult Ceratopogonidae specimens belonging to *Alluaudomyia, Atrichopogon, Bezzia, Culicoides, Dasyhelea, Echinohelea* and *Forcipomyia* genera were collected.

Of these seven genera, the most abundant were *Dasyhelea* (48.06%), followed by *Forcipomyia* (26.94%) and *Atrichopogon* (13.61%) which represented 90% of the total specimens, while the other percentage corresponded to *Culicoides* (10.56%), *Alluaudomyia, Echinohelea* and *Bezzia* (0.28%, respectively) (Table 1).

## Chi-squared analysis

When considering the total abundance of specimens per sampling site, a higher abundance was observed at El Corralito (EC, 40.83%), followed by Marapa Central (MC, 18.61%) and El Badén (EB, 17.78%). Yánima (YA) and BajoMarapa (BM) exhibited similar abundances (11.67% and 11.11%, respectively).

For the bivariate analyses, the four more abundant genera were considered. Significant differences among sampling sites, seasons and genera were observed (Tables 2 and 3).

*Dasyhelea* genus was the most abundant in autumn (April), winter (July) and summer (February), and at EC, EB, BM and MC sites; followed by *Forcipomyia*, with a higher abundance in spring (October) and at Yánima site. In turn, *Atrichopogon* was the most abundant genus in autumn (April), winter (July) and summer (March) at EC, EB and BM sites; while *Culicoides* was more abundant in spring (December) and at YA and MC sites. Finally, in all sampling sites, there was an increase in the abundance of the seven genera during the warmer seasons along the study period.

**Table 1** Absolute abundance of Ceratopogonidae during July, 2008–July, 2009, Juan Bautista Alberdi, Tucumán.

| Genera | El Corralito | El Badén | Yánima | Bajo Marapa | Marapa Central | Total | (%) |
|---|---|---|---|---|---|---|---|
| *Alluaudomyia* | 1 | 0 | 0 | 0 | 0 | 1 | 0.28 |
| *Atrichopogon* | 22 | 15 | 2 | 7 | 3 | 49 | 13.61 |
| *Bezzia* | 1 | 0 | 0 | 0 | 0 | 1 | 0.28 |
| *Culicoides* | 16 | 12 | 4 | 2 | 4 | 38 | 10.56 |
| *Dasyhelea* | 74 | 26 | 14 | 17 | 41 | 173 | 48.06 |
| *Echinohelea* | 1 | 0 | 0 | 0 | 0 | 1 | 0.28 |
| *Forcipomyia* | 32 | 11 | 22 | 14 | 19 | 97 | 26.94 |
| Total | 147 | 64 | 42 | 40 | 67 | 360 | 100 |

**Table 2** Chi-squared coefficient test table and V Cramer association coefficient for Ceratopogonidae, in relation to sampling sites and season.

| Rows × columns | Chi-squared | g.l | p-value | x of table | Coef. V of Cramer |
|---|---|---|---|---|---|
| Sities × genera | 38.92 | 12 | 0.0001 | 21.02 | 0.16 |
| Seasons × genera | 28.31 | 9 | 0.0008 | 16.91 | 0.14 |

**Table 3** Total abundance of the most common genera of Ceratopogonidae per seasonal and sampling site, Juan Bautista Alberdi, Tucumán.

| Sites | Season | *Atrichopogon* sp. | *Culicoides* sp. | *Dasyhelea* sp. | *Forcipomyia* sp. |
|---|---|---|---|---|---|
| 1 | Autumn | 1 | 0 | 13 | 5 |
| | Winter | 0 | 0 | 3 | 2 |
| | Spring | 1 | 3 | 2 | 1 |
| | Summer | 20 | 13 | 57 | 23 |
| 2 | Autumn | 0 | 1 | 3 | 1 |
| | Winter | 1 | 0 | 3 | 0 |
| | Spring | 1 | 1 | 1 | 2 |
| | Summer | 13 | 10 | 19 | 8 |
| 3 | Autumn | 2 | 1 | 2 | 0 |
| | Winter | 0 | 0 | 5 | 2 |
| | Spring | 0 | 0 | 0 | 1 |
| | Summer | 0 | 3 | 7 | 19 |
| 4 | Autumn | 1 | 0 | 3 | 5 |
| | Winter | 1 | 0 | 1 | 0 |
| | Spring | 2 | 1 | 0 | 2 |
| | Summer | 3 | 1 | 13 | 7 |
| 5 | Autumn | 0 | 0 | 6 | 2 |
| | Winter | 0 | 0 | 1 | 1 |
| | Spring | 0 | 2 | 1 | 11 |
| | Summer | 3 | 2 | 33 | 5 |
| | Total | 49 | 38 | 173 | 97 |

Note:
Site 1, El Corralito; Site 2, El Badén; Site 3, Yánima; Site 4, Bajo Marapa; Site 5, Marapa Central.

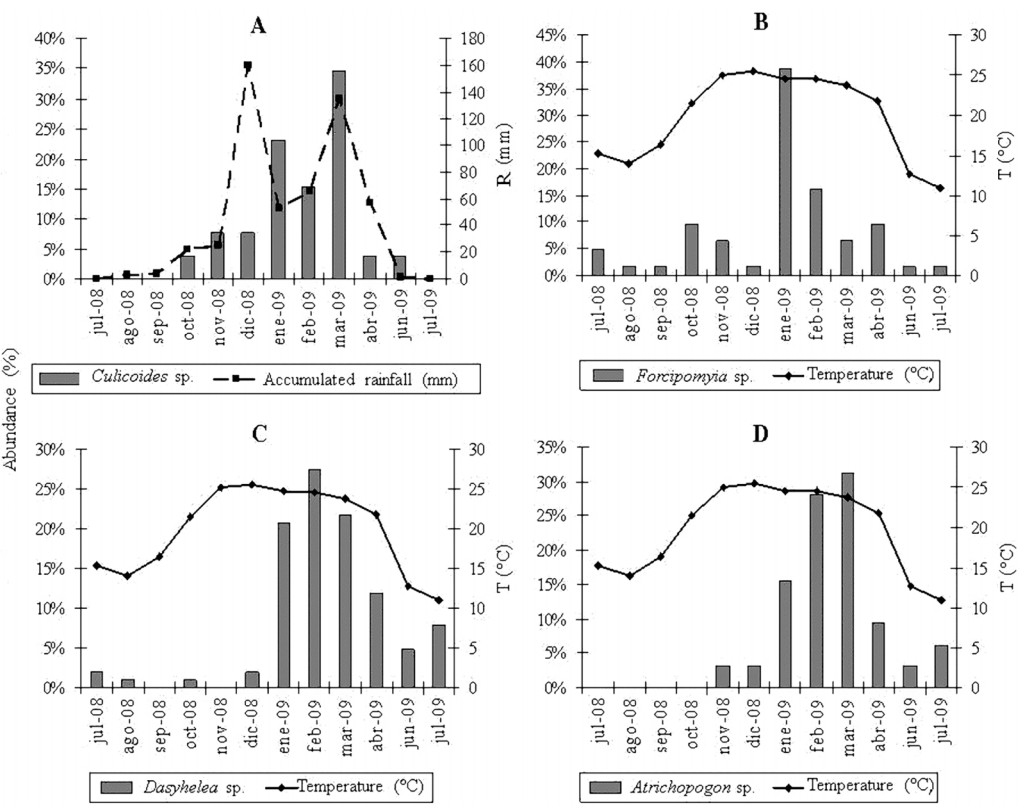

**Figure 2** Mean relative abundance of A. *Culicoides*, B. *Forcipomyia*, C. *Dasyhelea*, D. *Atrichopogon* versus regressor variables.

## Regression analyses

The multiple regression analyses allowed obtaining the following descriptive models: for *Culicoides* genus, the regression analysis between the abundance of specimens and meteorological variables determined a significant correlation with accumulated rainfall ($R^2 = 0.46$; $P < 0.0157$) (Fig. 2A); while temperature was strongly related to the abundance of *Forcipomyia* ($R^2 = 0.32$; $P < 0.0561$) (Fig. 2B), *Dasyhelea* ($R^2 = 0.59$; $P < 0.0035$) (Fig. 2C) and *Atrichopogon* ($R^2 = 0.42$; $P < 0.0221$) (Fig. 2D).

## Partial and predicted residuals

From the partial residuals (Figs. 3A–3D), a positive linear relation was observed between *Culicoides* abundance and accumulated rainfall, while the same relation was observed between temperature and the abundance of *Forcipomyia*, *Dasyhelea* and *Atrichopogon*, although less marked in the latter. In addition, standardized residuals versus predicted (Figs. 4A–4D) determined a dispersed point cloud, which indicated that the used model was valid for three of the four studied genera. Finally, the trend of the points of *Atrichopogon* was negative, indicating that the model was unsuitable for the regressor variable retained by the model.

## DISCUSSION

In the present study, the abundance of Ceratopogonidae genera in an area of Yungas which has been strongly modified and transformed by humans was registered.
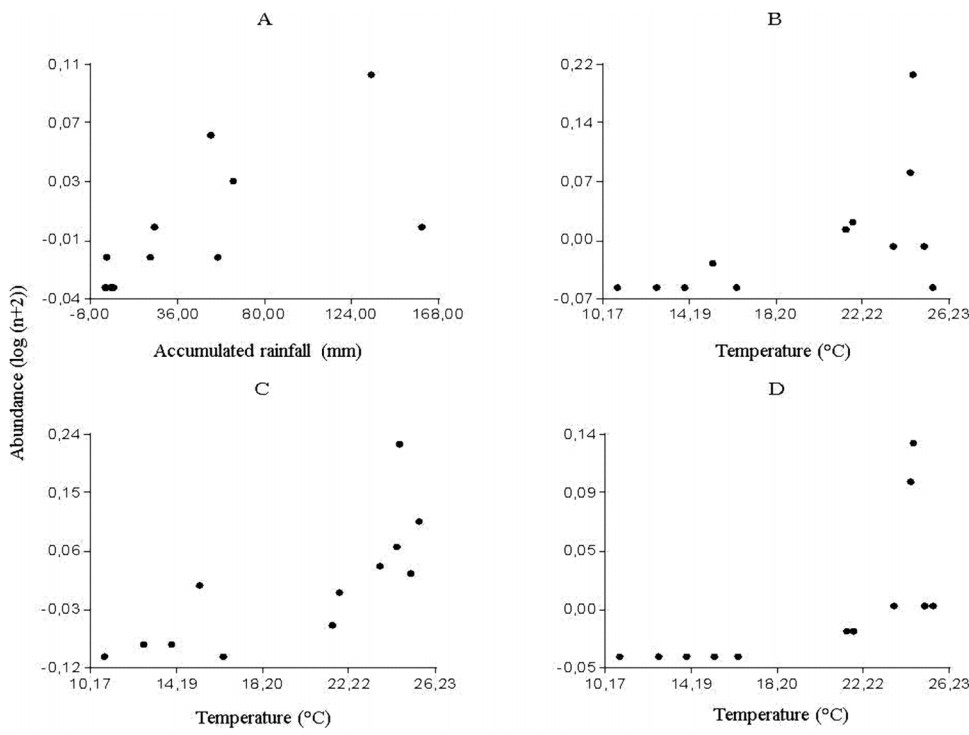

**Figure 3** Partial residuals of A. *Culicoides*, B. *Forcipomyia*, C. *Dasyhelea*, D. *Atrichopogon* and retained variables by the statistical model.

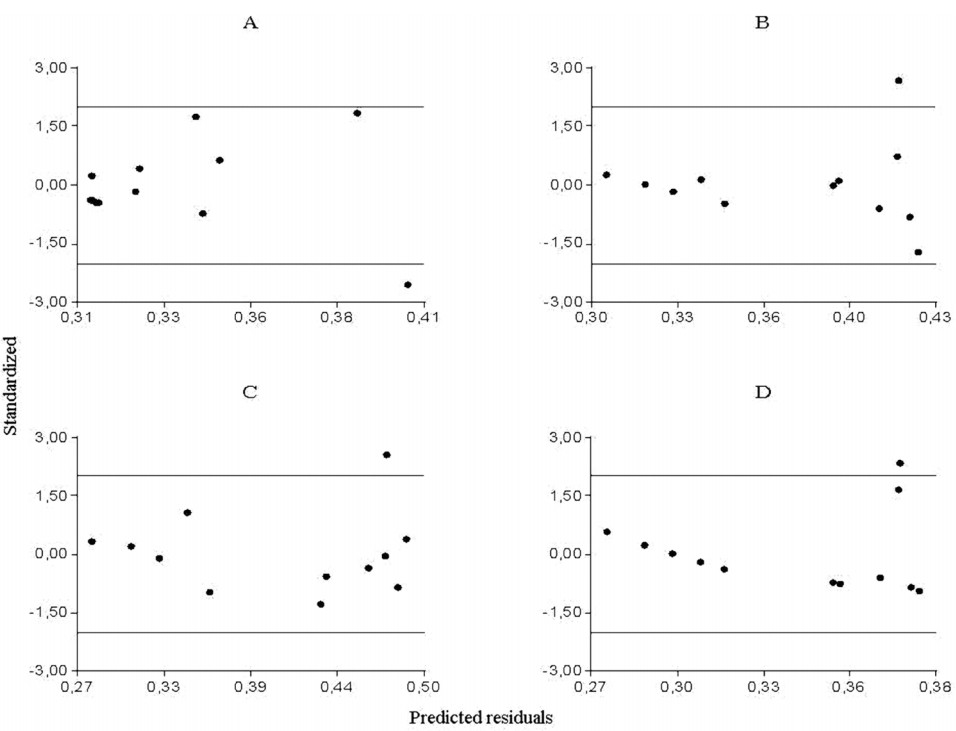

**Figure 4** Standardized vs. predicted residuals for A. *Culicoides*, B. *Forcipomyia*, C. *Dasyhelea*, D. *Atrichopogon*.

The abundance of specimens varied among seasons and study sites, with a differential pattern observed mainly in the warm season and at EC, MC, EB, YA and BM sites. Such differences in abundance could be due to that suggested by *Borkent & Spinelli (2007)*, who mentioned that most members of Ceratopogonidae family require of environments with high humidity for their development. For this family, the humid season is more suitable to complete their life cycle; since many of the genera go through winter in the later larval stage, similarly to what occurs at temperate regions of the north. Also, the available food sources in the study area, such as barnyard animals (chicken, pigs and horses) or humans in the case of hematophagous genera; and vegetal food sources such as flowers and fruits for pollinators should be considered; as well as the presence of suitable environments for the development of immature stages, which can be aquatic, semi-aquatic or terrestrial.

Regarding seasonality of *Culicoides* genera in northwestern Argentina, *Veggiani Aybar et al. (2010b)* and *Veggiani Aybar et al. (2012)* determined population peaks during summer, autumn and spring to a lesser extent in Tucumán province, and during spring and summer but gradually diminishing towards winter in Salta province; in agreement with in the findings of the present study.

Among the collected genera, *Culicoides* and *Forcipomyia* exhibit public health significance; the former excelling as the main vectors of *M. ozzardi* in the region. However, *Forcipomyia* genus (*Lasiohelea* subgenus), scarcely studied in the province, might be involved not only in the transmission of mansonelliasis but also in the transmission of other viruses and protozoa, which would represent a potential risk for the region, especially due to its high abundance found both in the present study as well as in other studies carried out in northwestern Argentina (*Veggiani Aybar et al., 2010a*; *Veggiani Aybar et al., 2015*). On the other hand, it is worth mentioning that in the last years, infection of *Culicoides* species with *Leishmania infantum* (LV) in the Old World was determined (*Seblova et al., 2012*; *Slama et al., 2014*), parasite that causes visceral leishmaniasis (VL) worldwide (*World Health Organization, 2010*). In the study area, *Salomón et al. (2006)* determined the spatial and temporal distribution of risk and the regional epidemiological trends of tegumentary leishmaniasis (TL), a type of *Leishmania* which is endemic of the Yungas of Argentina. The data presented here in addition to other studies carried out in the area (*Veggiani Aybar et al., 2010b*; *Veggiani Aybar et al., 2012*; *Veggiani-Aybar, 2015*) represent a starting point for continuing with the study of this genus in areas where its distribution matches that of the family Psychodidae, the main vectors of this parasite in the region (*Córdoba Lanús & Salomón, 2002*; *Salomón et al., 2006*; *Quintana, Fernández & Salomón, 2012*).

Regarding the influence of meteorological variables over the abundance of genera of medical and veterinarian importance, accumulated rainfall and temperature were significantly important for *Culicoides* and *Forcipomyia*, respectively. In relation to this, *Veggiani Aybar et al. (2010a)* and *Veggiani Aybar et al. (2011)* observed that in Tucumán province, the higher incidence of *Culicoides* was mainly associated with accumulated rainfall, followed by relative humidity, wind speed and mean temperature, although these last two variables were not significant in the present study. On the other hand, studies in

Salta province reported a positive correlation between the abundance of *Culicoides* and temperature and relative humidity (*Veggiani Aybar et al., 2012*). In addition, several authors have informed the strong relation between temperature, accumulated rainfall and humidity to the abundance of *Culicoides* in Brazil, due to the influence of these meteorological variables over the life cycles of the species or the alteration in their breeding sites (*Sherlock & Guitton, 1964*; *Santos Da Silva et al., 2001*; *De Barros, Marinho & Rebêlo, 2007*).

Studies assessing ecological aspects of *Forcipomyia*, *Dasyhelea* and *Atrichopogon*, in northwestern Argentina are scarce. However, *Veggiani Aybar et al. (2010a)* and *Veggiani-Aybar (2015)* reported the presence and abundance of these genera and the presence of *Brachypogon*, *Monohelea*, *Stilobezzia* and *Clinohelea* in the Yungas of Argentina and Bolivia. It is worth mentioning that their role as pollinators of economical important crops has not been evaluated in the region, although several studies in Latin America corroborate it (*Kaufmann, 1975*; *Young, 1983*; *Bravo, Somarriba & Arteaga, 2011*; *Córdoba et al., 2013*). Such a background highlights the importance of these genera, not only for natural ecosystems but also for agricultural systems of northwestern Argentina, where a wide variety of fruit crops which might be pollinated by these species is registered. Finally, *Bezzia*, *Echinohelea* and *Alluaudomyia*, which are predators of insects, might act as controllers of pests insects associated to crops.

The need of developing further research in other areas of northwestern Argentina strongly emerges from the results of this study, for upgrading the knowledge of both taxonomic and distributional aspects of Ceratopogonidae family, and for evaluating their relevance as disease vectors and pollinators of commercial crops.

## ACKNOWLEDGEMENTS

This study was conducted within the framework of a Federal Productive Innovation Project "controlled experimental intervention for the interruption of vector transmission of leishmaniasis in Tucumán endemo-epidemic area."

### Funding

This research was supported by the Proyectos Federales de Innovación Productiva PFIP 2006-1. The funders had no role in study design, data collection and analysis, decision to publish, or preparation of the manuscript.

### Grant Disclosures

The following grant information was disclosed by the authors:
Proyectos Federales de Innovación Productiva PFIP 2006-1.

### Competing Interests

The authors declare that they have no competing interests.

## Author Contributions

- José Manuel Direni Mancini conceived and designed the experiments, performed the experiments, analyzed the data, contributed reagents/materials/analysis tools, wrote the paper, prepared figures and/or tables, reviewed drafts of the paper.
- Cecilia Adriana Veggiani-Aybar contributed reagents/materials/analysis tools, wrote the paper, prepared figures and/or tables, reviewed drafts of the paper.
- Ana Denise Fuenzalida performed the experiments, analyzed the data, reviewed drafts of the paper.
- Mercedes Sara Lizarralde de Grosso reviewed drafts of the paper.
- María Gabriela Quintana conceived and designed the experiments, performed the experiments, analyzed the data, contributed reagents/materials/analysis tools, reviewed drafts of the paper.

## Data Deposition

The raw data has been supplied as Supplemental Dataset Files.

## Supplemental Information

Supplemental information for this article can be found online at http://dx.doi.org/10.7717/peerj.2655#supplemental-information.

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
