# Peer review of "Ceratopogonidae (Diptera: Nematocera) of the piedmont of the Yungas forests of Tucumán: ecology and distribution"

_PeerJ, doi:10.7717/peerj.2655_

## Round 0.1 · original submission · Minor Revisions

Please have the manuscript edited for English language usage prior to resubmission. In addition, a few other areas (in addition to those mentioned by reviewers) should be addressed.

It would be helpful to provide some description in the methods on how the selected sites were chosen and how they are similar or differ from one another. Are there any possible reasons why some areas would have a higher abundance than others?

The inclusion of the explanation of the linear regression can be abbreviated for the readers of this journal (line 125-139). A reference can be supplied for readers who wish to read up on this method in more detail.

In the Results (line 162) the reader is referred to Table 2 regarding observe differences among sites, seasons, and genera. However, Table 2 does not include any detail on seasonal differences. It would be helpful to include some more description on how seasonality was defined (what months constitute each season?), and to include a table showing data by site and season. This can be done for either most common genera, or for all genera collapsed.

Reviewer 1 ·

Basic reporting

In this article, the authors evaluate the presence and abundance of the main general of Ceratopogonidae in piedmont forest in Tucuman providence. They provide a good description of the genera under study and prepare the reader for the information to be provided. However, in their aims, the authors also want to determine the effect of meteorological variables in the distribution of the different genera but did not include enough information in the introduction regarding the current knowledge and the relevance of including this component in the study.

There are few spelling errors in the article that need to be fixed. For example where instead of were.

Experimental design

The methods section was clear and followed a logical process of data management including a strong data analysis plan.

Validity of the findings

Some sections of the results need to be revised. In the first part of the results, the numbers showed in the text do not match with the information provided. For example, the authors said that the most abundant genera were Forcipomyia with 26.86% followed by Culicoised (10.52%) but they found 47.92 of the specimens to be Dasyhelea.

Additional comments

In general, this is a good article but some minor changes are required.

Reviewer 2 ·

Basic reporting

OK

Experimental design

OK

Validity of the findings

OK

Additional comments

The wording should be revised and English. I am enclosing here the review (ms). Please, note that I am worked with track changes in the attached manuscript and comments.
It is important to note that the English is very basic. Although neither of us is a native speaker, we can notice that the English is poor, it fails in other sections (Abstrac, introduction,material & Methods, results, etc). We did our best making corrections, but we strongly suggest a well review by an expert in the language.

---

## Round 0.2 · Minor Revisions

Although the authors have corrected some of the English language issues in the manuscript, there are still language issues that should be addressed. Improvements in this area will greatly help the article, by improving the ability for readers to comprehend and cite it etc. We recommend that the authors perhaps find a native English language speaking colleague and/or an Editorial Services company to help. We appreciate the challenges of writing a scientific paper in a foreign language, as this is a task that most of us would not be able to accomplish on our own. In addition, please note the following specific comments (continuous line numbering was used):

Lines 109-113. The first sentence is not clear to me. If this is based on a previously used method, please provide the reference. Still not clear to me how the individual sites were selected. Some description of each of their specific characteristics would be helpful

Line 65: Specify type of equine encephalitis virus (i.e. WEE, VEE, etc…). References cited do not provide evidence of transmission of equine encephalitis viruses by Culicoides. Are authors referring to equine encephalosis virus?

Methods (data analysis): specify that this was multiple linear regression

Table 3: It would be easier to follow temporal trends at each sites if the rows were grouped by site. For example, the first four rows would be for site 1: autumn, winter, spring, summer

---

## Round 0.3 · Minor Revisions

Please note while the manuscript is close to acceptance, the English is still not up to the standard that is required. It appears that very few edits were made since the last submission. In addition, it appears that the editorial comment about the seemingly erroneous reference to equine encephalitis virus has not been addressed. Unless the authors can make the necessary improvements in the English language we will have no choice but to reject the submission. We suggest that you seek the input of an English speaking colleague to help edit the manuscript

---

## Round 0.4 · accepted · Accept

The English language is greatly improved. I noticed two small details, which I assume can be addressed during the productionstage:

1) There is a discrepancy in the % given for Atrichopogon in the abstract (13.85) versus Table 1 (13.61).
2) On line 168, I assume that the authors intend to say that the there was an increase in abundance during the warmer seasons, rather than "compared to" warmer seasons?